# Mechanistic Modeling of the Relative Biological Effectiveness of Boron Neutron Capture Therapy

**DOI:** 10.3390/cells9102302

**Published:** 2020-10-15

**Authors:** Seth W. Streitmatter, Robert D. Stewart, Gregory Moffitt, Tatjana Jevremovic

**Affiliations:** 1Medical Imaging Physics and Radiation Safety, Department of Radiology and Imaging Sciences, University of Utah Health, Salt Lake City, UT 84132, USA; 2Department of Radiation Oncology, University of Washington, Seattle, WA 98115, USA; trawets@uw.edu (R.D.S.); greg_moffitt24@msn.com (G.M.); 3International Atomic Energy Agency (IAEA), 1020 Wien, Austria; tatjana@chemeng.utah.edu

**Keywords:** RBE, BNCT, CBE, MCDS, RMF, MCNP

## Abstract

Accurate dosimetry and determination of the biological effectiveness of boron neutron capture therapy (BNCT) is challenging because of the mix of different types and energies of radiation at the cellular and subcellular levels. In this paper, we present a computational, multiscale system of models to better assess the relative biological effectiveness (RBE) and compound biological effectiveness (CBE) of several neutron sources as applied to BNCT using boronophenylalanine (BPA) and a potential monoclonal antibody (mAb) that targets HER-2-positive cells with Trastuzumab. The multiscale model is tested against published in vitro and in vivo measurements of cell survival with and without boron. The combined dosimetric and radiobiological model includes an analytical formulation that accounts for the type of neutron source, the tissue- or cancer-specific dose–response characteristics, and the microdistribution of boron. Tests of the model against results from published experiments with and without boron show good agreement between modeled and experimentally determined cell survival for neutrons alone and in combination with boron. The system of models developed in this work is potentially useful as an aid for the optimization and individualization of BNCT for HER-2-positive cancers, as well as other cancers, that can be targeted with mAb or a conventional BPA compound.

## 1. Introduction

Boron neutron capture therapy (BNCT) has been investigated as a potential treatment for glioblastoma multiforme (GBM), head and neck cancers, melanoma, and other tumor sites for many decades. Although efforts to develop neutron sources and new boron delivery agents for BNCT are ongoing [1], boronophenylalanine (BPA) and sodium borocaptate (BSH) are currently the only boron compounds approved for use in clinical trials [2]. Although doses of BPA are non-toxic at levels as high as 250 mg BPA/kg of body weight and tumor to blood ratios up to 3.5:1 [3], the use of non-conformal thermal and epithermal neutron beams is limited by the dose to normal brain tissue. Some clinical trials of BNCT with low-energy neutrons report confirmed radiation necrosis in non-tumor brain tissue [4]. Additionally, BSH is much more toxic than BPA and does not possess the specificity of BPA and is thus characterized as a global (non-specific) boron delivery agent [5]. For GBM, BSH was first used with thermal neutron beams in clinical trials started in Japan during the mid-1960s and then in the United States. Clinical trials of BNCT for the treatment of GBM ended in the United States in the early 1990s [6], although clinical trials of BNCT continued in other countries. Early studies of BNCT with non-conformal neutron beams ultimately concluded that BNCT using BSH is not superior to 3D conformal photon therapy in terms of patient survival [4]. Although BNCT with non-conformal neutron beams for the treatment of GBM has not proven advantageous when compared to current photon therapy, it has shown promise for the treatment of superficial melanoma lesions [7,8].

The optimal neutron spectrum that maximizes the dose from boron capture reactions has been extensively studied [9,10]. However, additional research on ^10^B pharmaceutical development is needed to further advance the overall use of BNCT for the treatment of cancer. Advances in tumor-specific boron delivery agents have the potential to greatly improve BNCT using non-conformal and conformal neutron beams. Conformal neutron therapy directs radiation to the beams-eye view of a tumor target, whereas non-conformal therapies deliver radiation to larger volumes of tissue without any tumor specificity in dose delivery. Some monoclonal antibodies (mAbs) are especially promising due to their high tumor selectivity. For example, Trastuzumab, an anti-HER-2 mAb, may prove to be an especially useful delivery agent for some cancers with overexpression of HER-2, specifically breast cancer [11,12]. This overexpression is present in 20–30% of breast cancer cases [13]. Current Positron Emission Tomography (PET) imaging studies show very good specificity of Trastuzumab, up to 18:1 tumor to healthy tissue ratio [14]. Although due to the size of the mAbs, diffusion is slow and optimal uptake of Trastuzumab, for example, occurs 3–5 days after injection [14].

In addition to having a highly specific neutron capture therapy (NCT) targeting compound, the use of highly conformal neutron beams that preferentially deliver much higher neutron doses to a tumor target than to nearby tissue and may help overcome the limits of earlier studies that delivered dose or boron to a tumor target with less specificity. The potential efficacy of BNCT using well-collimated, higher-energy neutron beams [15,16] is at an early stage of development, in part because very few facilities have the ability to deliver neutron beams shaped to the beams-eye view of an irregularly shaped tumor target. The University of Washington (UW) Clinical Neutron Therapy System (CNTS) is the only remaining operational facility in the U.S. with the ability to deliver 3D conformal fast neutron beams for the treatment of cancer [17,18,19].

In this work, we use a published Monte Carlo model for estimating the initial DNA damage arising from the interactions of photons, neutrons, and light ions **[20]** to model the initial distribution of DNA double-strand breaks (DSBs) arising in cells directly irradiated by neutrons and by the secondary charged particles arising from boron neutron capture reactions, i.e., BNCT. The mechanistic repair-misrepair-fixation (RMF) model [21,22] is then used to explicitly link DSB induction to estimates of the *α* and *β* parameters in the linear quadratic (LQ) cell survival model. Published studies of the measured cell surviving fraction for neutrons alone and in combination with boron are in good agreement with predictions from this system of Monte Carlo models. A MCNP (Monte Carlo N-Particle) model [19] of the University of Washington Clinical Neutron Therapy System (UW CNTS) and a few other neutron source models are used to explore the potential effectiveness of BNCT using BPA (boronophenylalanine) and a monoclonal antibody (mAb) that targets the HER-2 receptor. The impact of the microdistribution of ^10^B within and near representative cells on the relative biological effectiveness (RBE) and compound biological effectiveness (CBE) is examined. The results of these studies suggest that BNCT with fast, conformal neutron therapy beams may provide superior local tumor control compared to three-dimensional conformal neutron therapy alone or BNCT with non-conformal neutron sources.

## 2. Materials and Methods

### 2.1. Conceptual Aspects of a Multiscale Radiobiological Model 

Consider a small region of tissue or culture medium that receives a uniform absorbed dose *D* of ionizing radiation, as conceptually illustrated in Figure 1b. In ICRU Report 36 on Microdosimetry [23], the absorbed dose in a region of interest (ROI) is the product of the average event frequency ν times the frequency-mean specific energy, i.e., $D\equiv \nu {\bar{z}}_{F}≅\left(\Phi A\right){\bar{z}}_{F}$. Here, Φ is the particle fluence and *A* denotes the cross-sectional area of a subcellular, cellular or multicellular target of interest within the ROI. By definition, the absorbed dose distribution in a ROI may be considered uniform when, for any target within the ROI, the product $\nu {\bar{z}}_{F}$ or $\left(\Phi A\right){\bar{z}}_{F}$ is the same at all locations within the ROI. However, because of the stochastic nature of particle interactions within cellular and subcellular targets, the specific energy (stochastic analog to absorbed dose) imparted to different targets within a uniformly irradiated ROI may be quite different. That is, the absorbed dose in the ROI is the average (expected value) of the specific energy distribution of the cellular or subcellular targets within the ROI.

As a first approximation, the mean specific energy for a spherical target of diameter *d* irradiated by a charged particle of defined linear energy transfer (LET) randomly passing through a spherical target with mass density *ρ* is ${\bar{z}}_{F}≅LET/\rho d^{2}$ [23]. When a ROI is irradiated by a low-LET radiation, such as the energetic electrons arising from the interactions of ^60^Co γ-rays or MV X-rays, the mean specific energy per event is small (~1.6 mGy for a 0.2 keV/µm electron passing through a 5 µm in diameter target) and the number of events per unit absorbed dose is two or three orders of magnitude larger (ν ~600 for a 5 μm target and 0.2 keV/μm electrons). For higher LET radiations, such as the particles produced in BNCT (*n*, α) reactions, the mean specific energy increases in approximately linear fashion with particle LET, and the number of events per unit absorbed dose therefore decreases in linear fashion with increasing particle LET. The effects of low and high-LET radiations arising from stochastic differences in dose on the small scale even when the average absorbed dose on the larger (multicellular, macroscopic) scale is uniform motivates the definition of a radiation’s relative biological effectiveness (RBE). For two different types of radiations that result in the same biological effect *E*, the RBE of a radiation relative to another is defined as the absorbed dose of the (usually low-LET) reference radiation *D*_γ_ to the absorbed *D* of the other (usually higher LET) radiation, i.e., RBE ≡ *D*_γ_/*D*.

From the definition RBE, one can also define the RBE-weighted dose (RWD) as the product of (RBE × *D*). Conceptually, the RWD is the dose of the (usually higher LET) test radiation that produces the same biological effect as the reference radiation. For a ROI that receives a uniform absorbed dose of radiation from mixed particles, the overall RWD is the sum of (RBE*_i_* × *D**_i_*) integrated over all *i* particle types (charge and mass) and kinetic energies, i.e.,
(1)$$RWD=\sum_{i}\int_{0}^{\infty }dED_{i}\left(E\right)RBE_{i}\left(E\right)$$

The corresponding RBE, averaged over all particle types and energies, is
(2)$$RBE=\frac{1}{D}\sum_{i}\int_{0}^{\infty }dED_{i}\left(E\right)RBE_{i}\left(E\right),\;\mathrm{where}\;D\equiv \;\int_{0}^{\infty }dED_{i}\left(E\right)$$

Equations (1) and (2) provide a rigorous quantitative and conceptual framework to define a relevant RBE and RWD for one or more cells in a (macroscopic) ROI receiving a uniform absorbed dose of radiation. Conceptually, RBE*_i_*(*E*) is a biological dose–response function that primarily corrects for the small-scale, cellular, and multicellular ionization density (track structure) of the *i*th type of particle with kinetic energy *E*. For a non-uniform absorbed dose distribution (e.g., in vivo irradiation of tissue or a tumor), subdivide the ROI into a series of *j* smaller ROI that receive a uniform absorbed dose. Then, compute the overall dose-averaged RBE by summing the RWD over all *j* ROI and then dividing by the sum of the doses to all *j* regions, i.e.,
(3)$$RBE=\frac{1}{D}\sum_{j}RWD_{j},\;where\;D\equiv \sum_{j}D_{j}\;$$

Although the approach outlined by Equations (1)–(3) attributed the biological effectiveness of one type of radiation relative to another as arising from the small-scale (cellular and subcellular) ionization density (track structure), there is good evidence in the literature that cell-to-cell signaling (e.g., so-called bystander effects) and the interactions of cells with their environment (e.g., in vitro vs. in vivo environment) has a substantial impact on the dose–response characteristics of biological end points ranging from initial DNA damage to neoplastic transformation and cell death. At the tissue and organ level, immune responses and inflammation can also influence the clinically observable effects of radiation [24]. Mechanistic models that explicitly account for larger-scale tissue and tumor level of biology have not yet been proposed. However, the effects of cell-to-cell and other larger-scale biological effects are implicitly included the RBE model when the cell-, tumor- or tissue-specific parameters used in the model are adjusted to reflect measured in vitro or in vivo data.

In experimental determinations of radiation RBE, uncertainties in the dosimetry (e.g., non-uniform dose across a collection of cells in vitro or in vivo) as well as uncertainties in the measurement of a biological end point using a specific assay (e.g., *γ*-H2AX foci or PFGE for the measurement of DSB induction) contribute to uncertainties in RBE estimates. Uncertainties arise from random or systematic errors in the biological assay as well as random and systematic errors in the dosimetry. The dosimetry of low energy, very short range (high-LET and RBE) particles is especially challenging. It is not uncommon to have combined dosimetric and biological uncertainties in measurements that exceed 10%.

### 2.2. MCDS + MCNP Model for a Mixed Radiation Field

The Monte Carlo Damage Simulation (MCDS) algorithm, which simulates the induction and clustering of DNA lesions in anoxic to aerobic cells (O_2_ concentrations between 0–100%) uniformly irradiated by monoenergetic electrons, protons and particles up to ^56^Fe with energies as high as 1 GeV and for arbitrary mixtures of charged particles with the same or different kinetic energies, has been extensively tested and benchmarked against track structure simulations [25,26,27,28,29,30] and experimental data in previous work [31,32]. Details of the computationally efficient method used to integrate information from the MCDS into larger-scale MCNP simulations are described in Stewart et al. 2015 [20]. To apply the MCDS + MCNP system of models to a mixed radiation field, consisting of ions of varying charge, mass and kinetic energy, a standard MCNP energy deposition tally is modified by an ion-specific RBE_DSB_ dose–response function. The modified tally records the RWD averaged over a target region of interest. The dose-averaged value of the RBE_DSB_ is then computed by summing (RBE × *D*)*_i_* over all *i* ions and dividing by the total absorbed dose, as described by Equation (2). The dose-averaged values of ${\bar{z}}_{F}$ is obtained in the same manner for subsequent use in the model of cell survival. For MCNP simulations, the “tally precision stop” option was used to stop at the number of particle histories needed to reach a relative error (SEM) of 0.001 in the tally volumes. 

### 2.3. RMF Model for a Mixed Radiation Field

Within the RMF model [21,22], the effects of particle type and kinetic energy (and hence LET) on *α* and *β* in the linear quadratic (LQ) cell survival model are explicitly linked to the initial numbers and spatial distribution of DSB. In the RMF model, the biological processing of initial DSB into lethal chromosome aberrations or point mutations is modeled by a coupled system of non-linear differential equations. From combined MCDS + MCNP simulations, dose-weighted RBE_DSB_ and ${\bar{z}}_{F}$ are computed for each ion contribution and within the RMF, low and high-dose RBE (asymptotic limits) for the end point of reproductive cell death are computed [22,33] as
(4)$$RBE_{LD}=\frac{\alpha_{p}}{\alpha_{\gamma }}=RBE_{DSB}\left(1+\frac{2{\bar{z}}_{F}RBE_{DSB}}{{\left(\alpha /\beta \right)}_{\gamma }}\right),\;RBE_{HD}=\sqrt{\frac{\beta_{p}}{\beta_{\gamma }}}=RBE_{DSB}$$

These formulas are derived under the assumption that intra-track binary misrepair is negligible (≤1%) for the low-LET reference radiation, e.g., the cell-specific adjustable biological parameter *κ* = 2β_γ_/*∑*_γ_ and θ = α_γ_/*∑*_γ_ within the RMF (Figure 3A in [21]). Here, *∑*_γ_ is the DSB Gy^−1^ Gbp^−1^ for the reference radiation and *∑*_p_ is the DSB Gy^−1^ Gbp^−1^ for the test radiation, and hence, the ratio *∑*_p_/*∑*_γ_ = RBE_DSB_.
(5)$$RBE_{LD}=\frac{\alpha_{p}}{\alpha_{\gamma }}=\frac{\theta \Sigma_{p}+\kappa \Sigma_{p}^{2}{\bar{z}}_{F}}{\alpha_{\gamma }}=\frac{\frac{\alpha_{\gamma }}{\Sigma_{\gamma }}\Sigma_{p}+\frac{2\beta_{\gamma }}{\Sigma_{\gamma }^{2}}\Sigma_{p}^{2}{\bar{z}}_{F}}{\alpha_{\gamma }}=RBE_{DSB}\left(1+\frac{2{\bar{z}}_{F}RBE_{DSB}}{{\left(\alpha /\beta \right)}_{\gamma }}\right)$$
(6)$$RBE_{HD}=\sqrt{\frac{\beta_{p}}{\beta_{\gamma }}}=\sqrt{\frac{\frac{\kappa }{2}\Sigma_{p}^{2}}{\beta_{\gamma }}}=\sqrt{\frac{\frac{2\beta_{\gamma }}{2\Sigma_{\gamma }^{2}}\Sigma_{p}^{2}}{\beta_{\gamma }}}=RBE_{DSB}$$

For direct comparison to experimental cell survival, the two terms in Equation (4) are simply solved for the linear (*α_p_*) and quadratic (*β_p_*) variable of the test radiation. Here *α**_γ_* and *β_γ_* are the LQ parameters for the low-LET radiation (e.g.,—^60^Co *γ*-rays and 200–250 kVp X-rays) and *α_p_* and *β_p_* are the LQ parameters for the test radiation, in this case, the dose-weighted LQ parameters are for the combined ion contributions of the neutron or BNCT field. Because biophysically meaningful values of the RBE_DSB_ as well as ${\bar{z}}_{F}$ and (*α*/*β*)*_γ_* must be non-negative, Equation (4) implies that the RBE for reproductive cell death must fall within the range of values defined by RBE_DSB_ (minimum RBE) and RBE_LD_ (maximum RBE) for a given cell line or tissue. For a single large absorbed dose of radiation, as is typical with conventional BNCT [*D* > (*α*/*β*)*_γ_*], RBE_DSB_ is the more relevant metric, while a fractionated regime of smaller absorbed doses, as is seen in fast neutron therapy [D < (*α*/*β*)*_γ_*], RBE_LD_, is the more relevant metric. The RMF formulas in combination with first principle estimates of RBE_DSB_ have been shown to reproduce trends in cell survival for electrons, protons and other charged particles with an LET up to at least 100 to 200 keV/μm. This framework has been used to predict cell survival for the mixed radiation field encountered in helium ion therapy [34] and heavy ion therapy [35], proton and carbon ion therapy [22,36] and for X-rays [33], monoenergetic deuterons and alpha particles [21].

For the lithium recoil ions encountered in BNCT, with LET ~370–390 keV/μm and ranges ~4–5 μm, which is comparable to the size of the nucleus (see Table 1), the RMF may overestimate the level of cell killing compared to experimental data for particles with LET > ~100 keV/μm (Figure 4 in [21], Figure 1 in [36]). Currently within the RMF, all DSBs have equal chances of contributing to cell killing, regardless of their proximity to one another and cell killing is predicted continue to increase past ~100–200 keV/μm. The MCDS corrects for continuous slowing-down approximation (CSDA) range and changes in stopping power as particles pass through a cell nucleus 5 μm (default) in diameter for estimates of ${\bar{z}}_{F}$, where the equation ${\bar{z}}_{F}≅LET/\rho d^{2}$ will overestimate ${\bar{z}}_{F}$ in this special case. Here, LET is the LET in water (keV/μm), *ρ* is the density of the nucleus (1.0 g/cm^2^) and *d* is the diameter of the nucleus (~5 μm). Figure 2 shows the trends in relative DSBs per track and μm per DSB vs. (Z_eff_/β)^2^ for alpha particles and ^7^Li ions; ${\bar{z}}_{F}\;$ values used are calculated via MCDS. As (Z_eff_/β)^2^ increases each particle reaches a specific peak in DSB per track and minimum DSB spacing. For the alpha particles, this corresponds to a (*Z_eff_*/*β*)^2^ ~4500, with a corresponding LET ~200 keV/μm, which is comparable to what has been seen in experimental studies of RBE vs. LET. This may play an important role in the RBE predictions for the high-LET capture products in BNCT and is discussed further in Section 3.3. The range of alpha particle energies seen in BNCT, as denoted in Figure 2, precede the particle-specific peak, while the lithium ion energies occur at the peak and past, where the effectiveness starts to decline. However, this peak occurs at a higher LET than seen in experimental data, suggesting that the minimum DSB spacing is smaller than the threshold the cell “sees” for processing. The increase in spacing after the minimum in Figure 2 is an artifact of using the mean chord length in the calculation, rather than the CSDA range. 

### 2.4. Simulation of the Secondary Charged-Particle Spectrum

Neutrons undergo a number of interactions in soft tissue important to radiobiology. The dominant interaction for fast neutrons is (*n, p*) with hydrogen, while slow and thermal neutrons have a high probability of being captured via ^1^H(*n,* γ)^2^H and ^14^N(*n, p*)^14^C reactions. These are non-specific dose components that affect all irradiated tissue. In addition, there is the localized dose arising from ^10^B capture reactions that create short-range, high-LET alphas, and recoil ^7^Li nuclei. MCNPX [37] was used to track all the secondary ions within the water/tissue phantom and cellular model, including ^1^H, ^2^H, ^3^H, ^3^He, ^4^He^2+^, ^7^Li^3+^, and ions with Z > 2. To separate the ^7^Li^3+^ contribution from the rest of the heavy ions such as ^14^C, the special tally treatment entry FT RES 3007 in MCNPX was used. 

The neutron capture ion algorithm (NCIA) model was enabled by setting the 7th entry on the PHYS:N entry to 4 in MCNPX; this allowed for the production of ions from the n(^10^B, α)^7^Li reaction as well as enabling light ion recoil physics. This setting accounts for the ionization potential and uses the proper two-body kinematics to bank recoil particles with the proper energy and angle. Simulations were performed with a proton, alpha and heavy ion cutoff energy of 1 keV (lowest allowed by the code) and the Vavilov energy straggling model with the finest-allowed energy resolution in stopping power (efac = 0.99). CEM03 and LAQGSM models were selected over the default physics in the LCA entry, as recommended in the User’s Manual [37] and the neutron cross-section data used are primarily from ENDF/B-VII.0. Energy deposition tallies (F6) were set up for all charged particles of interest (all possible secondary ions from neutrons and photons) to determine the physical and biological dose. Additionally, modified F6 tallies were set up with dose response functions that relate particle energy to DNA and cellular damage, as discussed in Section 2.2 and Section 2.3.

### 2.5. Model for Cellular Dosimetry

In order to assess the microdosimetry of BNCT treatment and the impact of subcellular ^10^B distributions, a simple cell model was developed and parameters were evaluated using MCNPX. The cell model is shown in Figure 1b and consists of concentric spheres representing the cell cytoplasm and nucleus, which is a common approach in microdosimetry studies. Others [38] have shown that Monte Carlo simulation is a suitable method to assess the stochastic and heterogeneous nature of alpha particle and other heavy ion energy depositions. They show that MCNPX simulations of specific energy (*z*) deposited in the cell nucleus, the single-hit density of specific energy *f*_1_(*z*) and the mean specific energy ‹*z*_1_› were in good agreement when compared with the literature using simple geometry as small as 1 µm. 

Uniform ^10^B distributions, as well as heterogeneous distributions, which more realistically mimics clinically used BNCT pharmaceuticals, are assessed. Using data collected from experimental studies related to subcellular localization of BPA [39], ^10^B was incorporated into the representative cell compartments. ICRU brain tissue composition [40] was selected as representative of the composition of 9L rat gliosarcoma surrounding the cells in vivo. The extracellular matrix was modeled as a cube of tissue, with the cell model embedded in the center of the cube at a depth of 4 mm. The 4 mm size of the cube was selected to be larger than the CSDA of the charged particles of interest (Table 1) but small compared to the range of the incident neutron mean free path. The neutron source was modeled as a uniform, monodirectional disk source. For the sake of computational efficiency, Monte Carlo simulations were performed in two steps. First, neutrons from a disk source incident on a tissue phantom are scored along the central axis of the beam (Figure 1a). The tally of neutron spectrum at the depth of interest (1.5–1.7 cm) was used as the source in a second, microdosimetric simulation mimicking an in vitro experiment (as shown in Figure 1b). The secondary charged particle energy distribution and the DNA damage are based on tallies within the cell nucleus as the critical (sensitive) volume. The MCDS contains a subcellular dosimetry model for charged particles passing through water [32], while MCNP handles larger-scale dosimetry and accounts for any charged particle equilibrium (CPE) effects. Notice the divergence of the MCDS and analytic ICRU formula for $\;{\bar{z}}_{F}\;$ when the CSDA range approaches 5 μm or less, as shown in Table 1. 

### 2.6. Neutron Source Models for BNCT

To assess normal tissue RBEs and overall cell survival, the four neutron sources we considered are: The Massachusetts Institute of Technology Fission Convertor Beam (MIT-FCB) [10]. The MIT-FCB is a commonly used source for analyzing BNCT because of the purity and intensity of epithermal neutrons [9]. It has a very similar neutron spectrum and microdosimetric properties to the Brookhaven Medical Research Reactor (BMRR) [41,42].A compact neutron source or “neutron multiplier” source (NM source). This source uses a D-T reaction to generate neutrons [43]. All of the reported results are for this source are based on a published MCNPX model [43].A new CN source derived from the NM by removing the uranium sphere from the NM source.The UW CNTS [17,18,19,44,45,46], which uses 50.5 MeV protons incident on a Be target to produce a fast neutron energy spectrum.

The neutron fluence in the MIT-FCB source corresponds to a reactor power at 5 MW, which produces ~3 × 10^9^ n cm^−2^ s^−1^ epithermal (0.5 eV–10 keV) fluence [9]. Spectral data for this fission source were acquired from literature [47] without the need for additional modeling of the MIT-FCB beam. D-T sources were a major focus of interest due to their compactness, lower cost, and greater feasibility in a hospital setting than a reactor-based neutron source. Rasouli and Masoudi [43] proposed using a fissionable material as a neutron multiplier, effectively increasing the number of neutrons emitted from the D-T neutron generation. Their work built on the initial work of Verbeke et al. [48] on D-T and D-D neutron sources. The proposed beam shaping assembly (BSA) uses a combination of TiF_3_, Al_2_O_3_ as moderators, Pb as a reflector, Ni as a shield and Li-Poly (Lithiated Polyethylene) as collimation. This BSA combination was reproduced in MCNPX with two tally planes past the aperture to record the neutron spectrum and flux, as seen in Figure 3. Neutrons produced by D-T reaction of this source vary around 14.1 MeV by only ±7% [43], thus, it is assumed that neutrons are emitted isotropically and monoenergetically from the target in this model. 

#### MCNP6 Model of the Clinical Neutron Therapy System (CNTS)

While not a traditional neutron source for BNCT, the UW CNTS is the only remaining fast neutron therapy facility in clinical operational within the U.S. Currently, the CNTS is mainly used for palliative treatments of tumors refractory to photons and for selected head and neck cancers, including salivary gland tumors [45,46]. However, it may be feasible to further enhance the usefulness of fast neutron therapy by combining 3D conformal neutron therapy with BNCT [15,49,50,51] or by using fast, conformal BNCT in combination with 3D conformal and intensity modulated photon or proton therapy.

In the UW CNTS, fast neutrons are produced by 50.5 MeV protons incident on a 10.5 mm thick beryllium target, primarily through (*p*, *n*) and (*p*, *n* + *p*) reaction, but a small portion are created through (*p*, 2*n*), (*p*, 3*n*), and (*p*, *n* + α) reactions [19]. The incident proton beam was modeled in MCNP6 as a monoenergetic, monodirectional disk source of 0.5 cm radius, uniformly sampled. The beam originates in the vacuum above the beryllium target. Neutrons and photons are transported through the geometry as illustrated in Figure 4 and tallied in a volume of air below the target housing. The neutron spectrum and fluence at this point is recorded in a phase space file, using an SSW entry in MCNP6, and then transported as a secondary source through the multileaf collimator (MLC) (Figure 4b) and then into a water or tissue phantom. All of the simulations reported in this work are for an open 10.4 × 10.3 cm^2^ field at a depth of 1.7 cm in water, no wedge, small flattening filter, 148.5 cm source to surface distance or SSD. 

## 3. Results

### 3.1. Energy Fluence of the MIT-FCB, NM, CN and UW CNTS Neutron Sources

Figure 5 shows a comparison of the neutron energy fluence for the MIT-FCB, NM, CN and UW CNTS sources. The NM source (configuration d with a Ni shield and Li-Poly collimator) can produce fluence as high as 5 × 10^12^ n/s at the target, with a resulting epithermal fluence rate of ~4 × 10^8^ n cm^−2^ s^−1^ at the beam port (tally planes). This configuration was chosen due to its maximum epithermal flux compared to other material combinations [43]. With the removal of the uranium sphere (i.e., CN source), the neutron fluence rate decreases to ~2 × 10^8^ n cm^−2^ s^−1^. In the UW CNTS, an open 10.4 × 10.3 cm^2^ field (small filter, 148.5 cm SSD) produces a neutron fluence rate along the central axis of the beam at a depth of 1.5 cm in water of 1.91 × 10^8^ n cm^−2^ s^−1^, which corresponds to an absorbed dose rate in water of 60 cGy min^−1^ at the depth of maximum dose (1.7 cm). The fluence-averaged neutron energy for the MIT-FCB, NM, CN and UW CNTS sources are 11.0 keV, 0.46 MeV, 0.36 MeV, and 21.0 MeV, respectively. The average energy of the CNTS neutron energy spectrum varies with depth and lateral position within the field because of beam hardening as well as in-field and out-of-field nuclear interactions. As illustrated in Figure 5, all of the sources produce large numbers of thermal and epithermal neutrons. The NM and CM sources produce nearly identical neutron energy spectra over the entire energy range. Below approximately 20–30 keV, the MIT-FCB source also produces a neutron energy spectrum very similar to the NM and CN sources; however, the MIT-FCB source has been optimized to reduce the number of higher-energy neutrons. Unlike the MIT-FCB, NM, and CN sources, the UW CNTS beam also produces substantial numbers of very energetic neutrons (>10 MeV), which is advantageous for the delivery of a conformal neutron dose to tumor targets (i.e., MLC are used to shape the field to the beam’s eye tumor contour) but does little to enhance ^10^B(*n*, α)^7^Li reactions.

### 3.2. Proton and Alpha Particle Cell Survival Benchmarks

To test the accuracy of the proposed system of models, it is applied to experimental cell survival data for monoenergetic protons and alpha particles. The work of Goodhead et al. [52] (V79-4, HeLa human and C3H 10 T_1/2_ mouse cell lines in a variety of media in monolayers on custom Hostaphan-based dishes) compared cell survival of alpha particles and protons of equal LET, finding that protons had a significantly greater linear term in the dose–response for V79-4 cell inactivation as shown in Table 2 (α parameter equal to 0.42 Gy^−1^ vs. 0.25 Gy^−1^). They concluded that this must be due to differences in track structure. Table 2 shows the experimental results against model estimates, confirming that the track structure level effects are reflected in our system of models and not based on LET alone. At 1.4 MeV, the experimentally-derived value of α is larger than expected compared to the other experimental data and MCDS + RMF estimates. At 0.42 Gy^−1^, it is significantly larger than the α estimate for a 1.2 MeV alpha particle of 0.30 Gy^−1^. This is likely due to the experimentally uncertainty inherent in the dosimetry and cell counting statistical variations for these short-range, high-LET particles. It is expected that the α values for the 1.2 and 1.4 MeV alpha particle will only differ by a small amount since their (Z_eff_/β)^2^ values are similar. 

Additional tests of the model were performed for a range of alpha particle kinetic energies. In the work of Tracy et al. [53] cell survival in the V79-4 cell line (Chinese hamster cells recovered from the MRC stock in liquid N_2_ storage thawed and grown in T75 Eagle’s minimum essential media with 10% fetal calf serum and penicillian/streptomycin) was assessed for alpha energies from 1.1 to 4 MeV, which covers the energy range seen in ^10^B capture reactions. Figure 6 shows the comparison of experimental cell survival results and model estimates for the alpha particle energies examined. Table 3 compares the major radiobiology variables derived from the experimental work of Tracy et al. and estimates derived from our system of models. For high-LET particles, such as alpha particles in this range, note that MCDS + MCNP estimates of LET are very good for higher energy, longer range but start to diverge for the lower energies, where path length straggling and LET variations in the ion track come into play. In experimental irradiation conditions, truly monoenergetic beams are rarely achieved; there is at least some spread in the particle energy. Monoenergetic simulations were compared to simulations of the reported energy distributions, finding that the impact on RBE_DSB_ was >0.5%, while the impact on ${\bar{z}}_{F}$ was relatively large (4–14%) for the 1.1–1.8 MeV alpha particles, but <2% for the 2.4–4 MeV alpha particle energy distributions. Since Tracy et al. [53] reports a distribution of cell sizes in their cell survival experiments, nucleus diameters of 3–6 µm were assessed in MCDS + MCNP simulation. This variable (ndia) has a quite large impact on estimates of ${\bar{z}}_{F}$, but a small impact on RBE_DSB_ within the current version of MCDS. Table 3 shows that there is good agreement between MCDS + MCNP estimates of ${\bar{z}}_{F}$ and the computed values ${\bar{z}}_{F}$ from RMF fits to the experimental data (using Equation (4), RBE_LD_) for 1.1 and 1.5 MeV alpha particles, but exhibiting opposite trends at higher energy, opposite of the LET comparisons. This finding indicates that the accuracy of the RBE_DSB_ estimates or some other aspect of the RMF model may need to be refined in order to improve the accuracy of the model for very low energy (short-range, high-LET) alpha particles. Other work [34] supports the hypothesis that the RBE for cell survival of alpha particles can be reliably estimated within the RMF for clinically relevant scenarios in helium ion radiotherapy.

### 3.3. RBE of Selected Ions Produced in BNCT Reactions

Table 4 lists estimates of the dose-averaged RBE_DSB_ and RBE_LD_ for selected ions in BNCT reactions. All of the results in this table are based on a representative ^10^B subcellular distribution of 40 ppm in the cell cytoplasm. It can be seen that the recoil protons from ^14^N capture and the fast recoil protons from hydrogen elastic interactions with fast neutrons are lumped together in the single RBE value, but weighted appropriately by absorbed dose. The ^14^N content of the tissue can have a significant effect on the dose-weighted proton RBE because the capture reactions release higher RBE protons than the protons from hydrogen scattering. One of the more striking aspects of the results shown in Table 4 is that the proton low dose RBE with α/β = 3 Gy is not much higher than RBE_HD_ ≅ RBE_DSB_ whereas RBE_LD_ is much larger than RBE_HD_ for particles with *Z* > 2. These effects arise in the RMF model because intra-track DSB interactions (also referred to as “proximity effects” in the literature) are much more significant for ions with Z > 2 than for protons. In terms of Equation (4), the product of $2{\bar{z}}_{F}RBE_{DSB}/{\left(\alpha /\beta \right)}_{\gamma }$ is ≤ 1 for protons (with kinetic energies above 1 keV) and large (compared to unity) for heavier ions for α/β = 3 Gy. The results from Table 4 also suggest that the effects of α/β on the overall RBE_LD_ arise in the RMF model from intra-track (proximity) effects associated with heavier ions from BNCT reactions rather than protons.

### 3.4. In Vitro and In Vivo Testing of the Dosimetry and CBE Models

To calibrate the RMF model for BNCT, we first obtained the LQ parameters α_γ_ and β_γ_ for the 9L rat gliosarcoma cell line (in vitro and intracellular tumor cells maintained in DMEM medium supplemented with inactivated 5% fetal bovine serum) irradiated by 200–250 kVp X-rays, which was published in Coderre et al. [54]. It follows from Equation (4) that the only further parameters needed to estimate α_p_ and β_p_ for the BNCT experiment are ${\bar{z}}_{F}$ and RBE_DSB_, which are calculated with the MCDS and integrated into MCNP to produce dose-averaged values. The ^10^B concentrations of 40 ppm and 27 ppm for in vivo/in vitro and in vitro experiments, respectively, were modeled according to the Coderre et al. data [54]. For the in vivo/in vitro experiments, ^10^B was distributed primarily outside the nucleus (~100:1 cytoplasm to nucleus concentration) and for the in vitro experiment, it was homogenously distributed throughout the cell compartments, according to the findings of Nguyen et al. [39] (9 L gliosarcoma cells maintained in alpha medium supplemented with 10% fetal bovine serum and antibiotics) and Bennett et al. [55] (GS-9L rat glioma cell line from a tumor induced by N-nitrosourea and maintained in DMEM supplemented with 10% fetal bovine serum). As illustrated in Figure 7, estimates of the surviving fraction for the BNCT experiments of Coderre [54] with BPA in vivo/in vitro agree within 5% (solid red line) and neutron-only cell survival estimates for both the in vivo/in vitro, as well as in vitro experiments (solid blue lines) are also in good agreement. This provides some measure of confidence that the model may also be useful for predicting the photon-isoeffective doses for other neutron sources and known boron distributions. For comparison to the neutron source used by Coderre, estimates of cell survival for the CN, NM and CNTS sources with the same concentration of ^10^B are also shown in Figure 7. Estimates of cell survival are slightly higher for the CN, NM, and CNTS sources than for the source used by Coderre et al. [54].

For simplicity and uniformity, the dose-weighted RBE values in Table 3 use the RBE_LD_ formulation (Equation (4)) with ^60^Co as the reference radiation. However, if a different low-LET reference radiation is desired, a correction factor can be applied (e.g., 1.1 for 250 kVp X-rays, 1.3 mm Cu filtration) [20]. For fraction sizes that are small compared to α/β which encompasses the most clinically relevant range of doses used in fast neutron therapy (~1 Gy per day to a total as high as 16 or 18 Gy), RBE_LD_ is the relevant metric and is always ≥ RBE_HD_ (RBE_DSB_). In past clinical trials of BNCT treatment, a photon-equivalent dose of approximately 50 Gy is typically delivered in a single fraction [3]. Estimates of the photon-isoeffective dose based on the RBE_LD_ are a more relevant metric of the potential effectiveness of a fractionated BNCT treatment with fast neutrons. For a single acute dose or hypofractionated BNCT, estimates of the photon-isoeffective dose based on RBE_HD_ ≅ RBE_DSB_ is the more relevant metric of potential treatment effectiveness.

The cumulative RBE estimates are based on a cytoplasmic ^10^B concentration of 40 ppm for BPA and 100 ppm for mAb. The (α/β)_γ_ values of 87 Gy, 3 Gy, and 10 Gy are for the 9L rat gliosarcoma cell line (*in vivo*/*in vitro*), mammary carcinoma and a typical early responding tissue or tumor, respectively. The high-dose RBE (RBE_DSB_) is effectively the same for the NM, CN, and UW CNTS neutrons and slightly larger for the MIT-FCB neutrons, which supports the idea that the MIT-FCB source produces a secondary charged particle energy distribution with a closer to optimal LET distribution (see Table 5). The same general trends hold for the reproductive cell death in the limit when the dose per fraction is small compared to α/β (RBE_LD_). However, the models predict that the low-dose RBE will always be greater than or equal to the high-dose RBE. Further, the RBE_LD_ is predicted to increase with decreasing α/β. For the lower energy, MIT-FCB, NM, and CN neutron sources, RBE_LD_ is predicted to be the same as RBE_HD_ (~3) for all tumor or tissue types with α/β above 10 Gy; RBE_LD_ is also approximately equal to 3 for the UW CNTS with α/β = 87 Gy. For tumors or tissue with α/β = 3 Gy, RBE_LD_ may be as large as 4.3 for the NM beam or 7.4 for the UW CNTS beam. These observations suggest that BNCT may be most effective for the treatment of tumors with a low α/β ratio, such as tumors of the breast and prostate. However, with the UW CNTS beam, RBE_LD_ is ~4 even when α/β = 10 Gy. Fractionated BNCT treatments using the UW CNTS may be a very effective treatment even for tumors with larger α/β, especially since beams can be directed towards the patient and tumor targets from any direction (i.e., any couch position and gantry angles) and shaped to the beams eye view of the tumor using 40 individually movable leaves. The CNTS offers a degree of dose conformity not possible with thermal and epithermal neutron sources traditionally used for BNCT and the combination of dose escalation and conformity should prove advantageous. Figure 8, Figure 9, Figure 10 and Figure 11 illustrate the potential advantage the CNTS has for deep seated tumors, using fractionation (RBE_LD_) over epithermal beams. The use of a mAb as the boron carrier instead of BPA could also offer some modest increases in the potential effectiveness of BNCT [11,12] and better quantification of uptake, and hence, tumor to healthy tissue ratios using immuno-PET [56]. However, additional experimental work is still needed to confirm that mAbs can be an effective and targeted boron carrier. 

Currently, RBE and CBE weighting factors and isoeffective dose calculations derived from cell survival experiments have been universally applied to calculate biologically equivalent dose for BNCT clinical trials and treatment on human subjects. This involves many assumptions that have mainly been derived from non-human experiments ([57] and references therein). Although we assume boron concentration and biodistribution based on experimental data, the method put forth here offers a mechanistic prediction of biological weighting factors, LQ parameters and hence, isoeffective doses, based on the specific tissue and end point of interest.

## 4. Discussion and Conclusions

A system of dosimetry and radiobiological models is presented to predict RBE, CBE, and other important biological metrics for selected neutron sources, tissue types, and boron distributions. With only the (α/β)_γ_ from the reference radiation, RBE_DSB_ and ${\bar{z}}_{F}$ (which are estimation from first principles), as *ad hoc* biological (input) parameters, the presented BNCT model accurately predicts the cell survival for in vitro and in vivo/in vitro experiments with the neutron beam alone and with BPA to within a few percent (Figure 7). Applying the model to a hypothetical mAb boron carrier that targets HER-2+ cells, even conservatively assuming no localization in the cell nucleus, shows a significant increase in CBE. Compounded with the macroscopic advantage of having a higher tumor to healthy tissue ratio of ^10^B, this methodology shows promising applications for other, theoretical, or in development, boron carrier pharmaceuticals. However, the strength of the estimates from the system of models is ultimately limited by the accuracy of the experimental determination of the ^10^B subcellular distribution and α_γ_ and β_γ_. Although the predicted cumulative RBE values for the compact, D-T produced neutron sources and the fast neutron source are less than that for the MIT-FCB neutron source, evidence shows the high tumor uptake and high tumor to healthy tissue ratios achievable with the proposed pharmaceutical [14], which has the potential to overcome the fluence and CBE restraints seen with compact neutron sources. The results suggest that BNCT with fast, conformal neutron therapy beams should provide superior local tumor control compared to 3D conformal neutron therapy alone or BNCT with non-conformal neutron sources. In addition to the increased dose conformity and uniformity the CNTS can achieve, the differences in RBE_LD_ and RBE_HD_ can be exploited to increase the therapeutic ratio of BNCT treatments. The primary limitation is patient tolerance to repeated administration of BPA or another pharmaceutical.

The most compelling argument for applying our system of models to BNCT is the ease of implementation and the minimal number of adjustable parameters. In the RMF model, the cell-, tumor- and tissue-specific kinetics and fidelity of DSB repair are contained solely in α_γ_ and β_γ_, the low-LET experimentally-derived LQ parameters, and dose-weighted values of RBE_DSB_ and ${\bar{z}}_{F}$, which are obtained from the MCDS + MCNP simulations, are needed to estimate the α_p_ and β_p_ of all the ion components in the mixed field and the subsequent values of RBE_LD_ and RBE_HD_. As described above, this approach has been successfully applied for other mixed fields of light and heavy ions. Mairina et al. [34] concluded that the RMF framework was a good candidate for predicting cell survival with He ion beams, especially considering that its implementation only required α_γ_/β_γ_ as input, without requiring tuning and adjustment with other light ion cell survival data [34,58]. However, from our investigation of the low-energy, high-LET alpha particles and previous work [21,22], there is evidence that refinements are needed for this subset of particles. Proximity effects, discussed earlier, which are not explicitly considered in the RMF, likely have an increasing importance as charged particles reach very high-LET. In the case of fast neutron therapy or boron neutron capture enhanced fast neutron therapy, this overestimation will likely not have a significant impact on RBE estimates, considering the other uncertainties in biological parameters.

Horiguchi et al. [59] used the particle transport simulation code (PHITS) coupled with the microdosimetric kinetic model (MKM) to estimate the relative biological effectiveness factors for BNCT. Within the MKM, cell survival is estimated from the probability densities of specific energies in a subcellular structure contained in the cell nucleus (domain). This adds at least one additional adjustable parameter as compared to the RMF, where the entire nucleus is considered. Additionally, as compared to our method, the PHITS + MKM model simulated the four BNCT dose components separately, where we obtained the biophysical variables for all components in one simulation. Subsequent fitting and optimization was also needed to update the domain radius. This framework has also been used to estimate biological dose and cell survival fraction in charged particle therapy [60,61].

Gonzalez and Santa Cruz [62] proposed a method to calculate the photon-isoeffective dose in BNCT to replace the old paradigm of using “RBE-weighted” doses for calculating the photon-equivalent dose. They show that using the fixed-RBE approach is not suitable to understand the observed clinical results in terms of the photon radiotherapy data and always predicted much higher equivalent doses that the isoeffective approach. They use a modified linear quadratic (MLQ) model to account for synergistic effects between low and high-LET components (i.e., sublesions produced by one radiation can combine with the sublesions produced by any other radiation to form lethal lesions). While not explicitly shown in the RMF equations, synergistic (inter-track and intra-track) DSB interactions are embedded in the RBE_DSB_ and RBE_DSB_ × ${\bar{z}}_{F}$ terms, with the RBE_DSB_ (relative DSB Gy^−1^ Gbp^−1^) and RBE_DSB_ × ${\bar{z}}_{F}$ (relative DSB track^−1^ Gbp^−1^) representing the intra-track and inter-track (proximity) effects, respectively. Within the RMF, the RBE_HD_ is only dependent on the dose-averaged RBE_DSB_, making it straightforward to implement as compared to Equation (15) in Gonzalez and Santa Cruz [62]. Further, note that the LQ parameters in MLQ were obtained from fitting of experimental data, requiring at least four variables. The range of survival fractions, S(D), and isoeffective doses, DR(D), can be obtained with some simple rearrangement of the RMF formulas.

Additionally, optimized target [15] and filtration of the UW CNTS and the advantage of the more conformal neutron beam have not been taken into consideration here, but may very well show promise for BNCT applications. Current applications using BPA for tumor treatments other than GBM (e.g., melanoma) may also benefit from more accurate RBE models.

## Figures and Tables

**Figure 1 cells-09-02302-f001:**
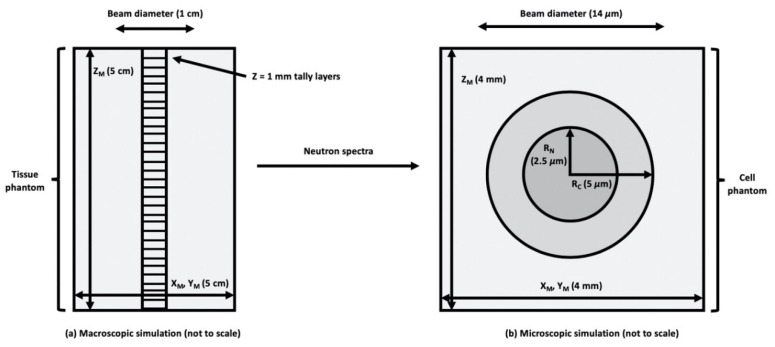
MCNPX (Monte Carlo N-Particle eXtended) models for: (**a**) water and tissue phantom (macroscopic simulation), and (**b**) cellular microdosimetry (microscopic simulation). Z_M_ = z dimension of medium, X_M_ = x dimension of medium, Y_M_ = y dimension of medium, R_N_ = nucleus radius, and R_C_ = cytoplasm radius.

**Figure 2 cells-09-02302-f002:**
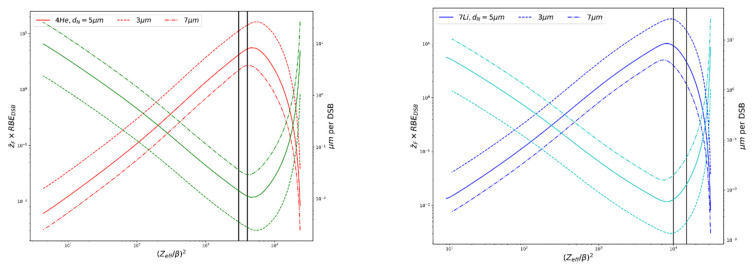
Plot of ${\bar{z}}_{F}\;\times \;$ RBE_DSB_ (relative DSBs per track), red and dark blue, and μm per DSB, green and light blue, vs. (Z_eff_/β)^2^ for alpha particles and ^7^Li ions, illustrating the particle-specific maximum DSB per track and minimum distance between DSBs, formulas used are described in [20]. *d* = nucleus (target) diameter. Vertical lines bracket the ion energies relevant to BNCT.

**Figure 3 cells-09-02302-f003:**
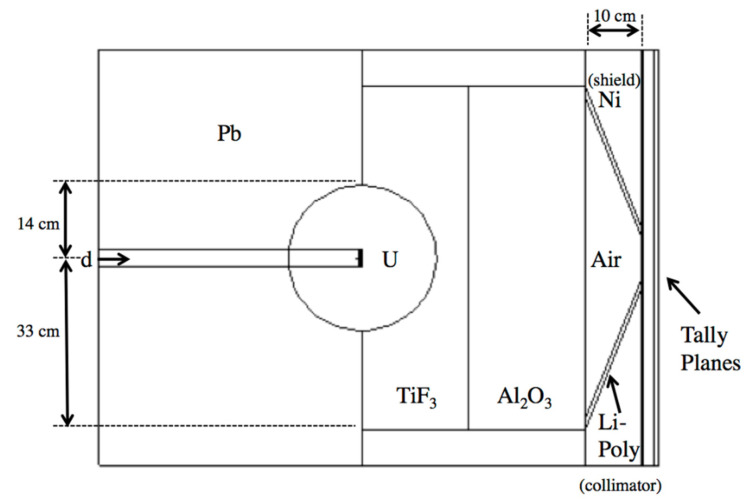
MCNPX model of the neutron multiplier and beam shaping assembly as proposed by Rasouli and Masoudi [43].

**Figure 4 cells-09-02302-f004:**
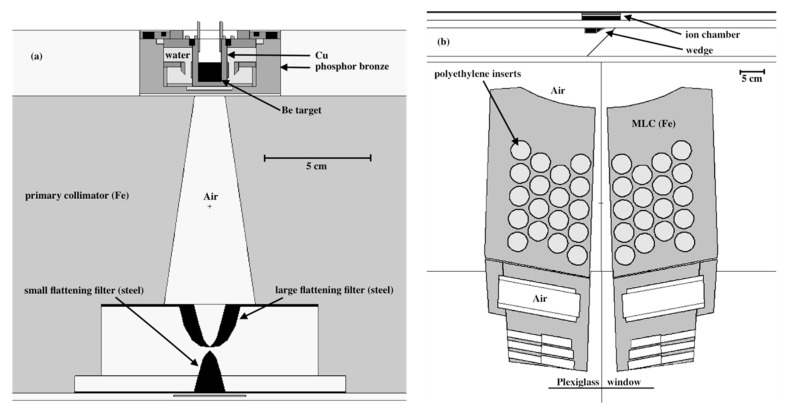
UW CNTS treatment head, (**a**) primary neutron production and collimation, and (**b**) the MLC downstream from (**a**). Additional details and benchmarks of the MCNP6 model of the CNTS are described in Moffitt et al. [19].

**Figure 5 cells-09-02302-f005:**
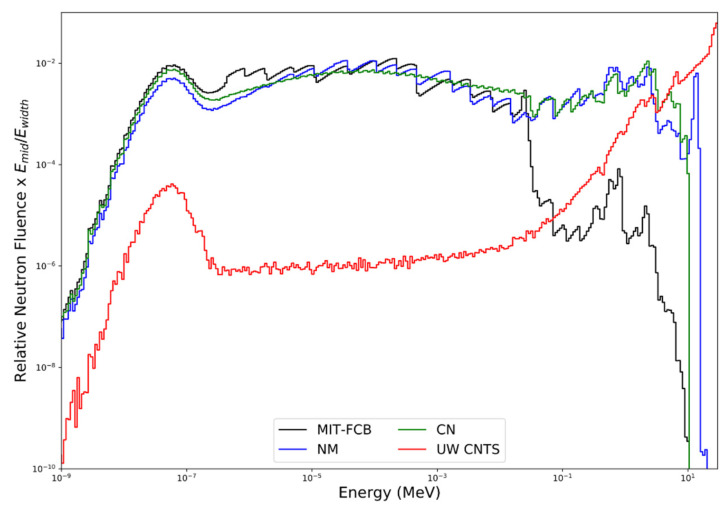
Comparison of the MIT-FCB, NM, CN and UW CNTS relative neutron fluence at a depth of 1.5 cm in a water phantom (geometry in Figure 1) obtained using MCNPX.

**Figure 6 cells-09-02302-f006:**
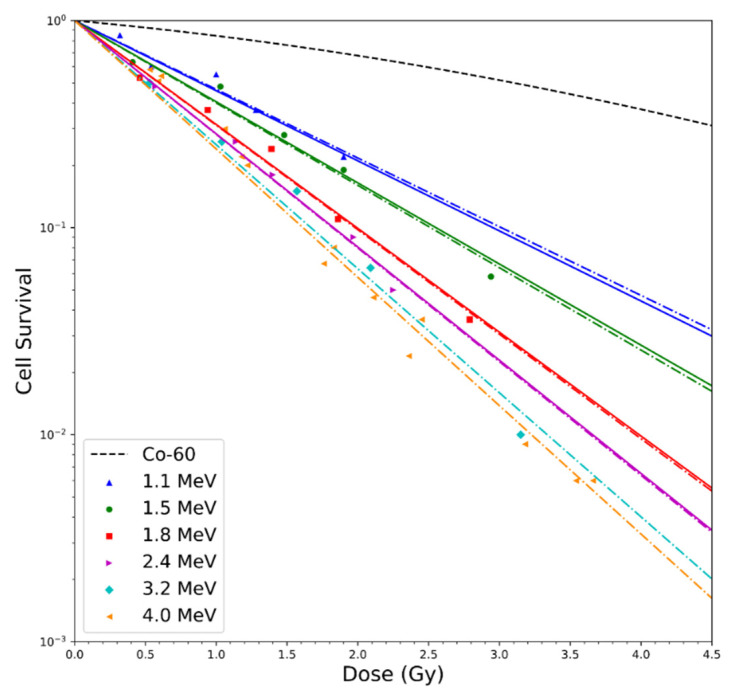
Comparison of cell survival in V79-4 cells irradiated by low-energy alpha particles [53]. Dashed lines are LQ fits to the experimental data and solid lines are RMF estimates. For 1.1 and 1.5 MeV (blue and red lines), ${\bar{z}}_{F}$ is obtained from MCDS + MCNP simulations, for 1.8 and 2.4 MeV (red and magenta lines), ${\bar{z}}_{F}$ is obtained from an RMF fit.

**Figure 7 cells-09-02302-f007:**
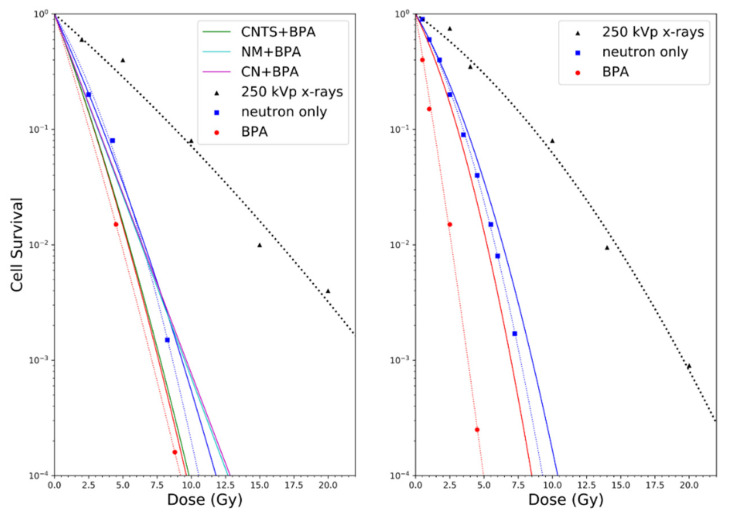
Cell survival predictions for the 9L rat gliosarcoma cell line with BPA and varying neutron source (parameter details in Section 3.3), in vivo/in vitro (**left**) and in vitro (**right**). Dotted lines are LQ fits, solid lines are RMF predictions.

**Figure 8 cells-09-02302-f008:**
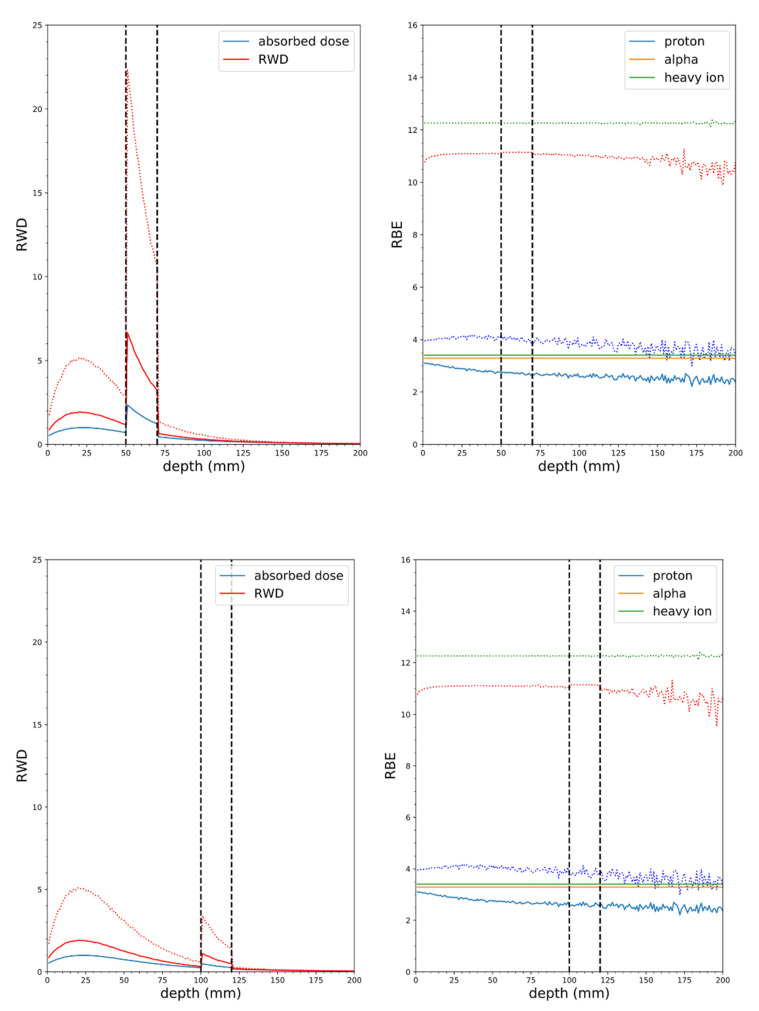
Soft tissue phantom simulations with the MIT-FCB neutron source, mass at 5 cm depth (top panel), 2 cm wide, 10:1 tumor to tissue ratio, 100 ppm ^10^B in tumor (vertical dashed lines indicate tumor borders), solid lines RBE(DSB), dotted lines RBE_LD_ (α/β = 3 Gy). Tumor at 10 cm depth (bottom panel).

**Figure 9 cells-09-02302-f009:**
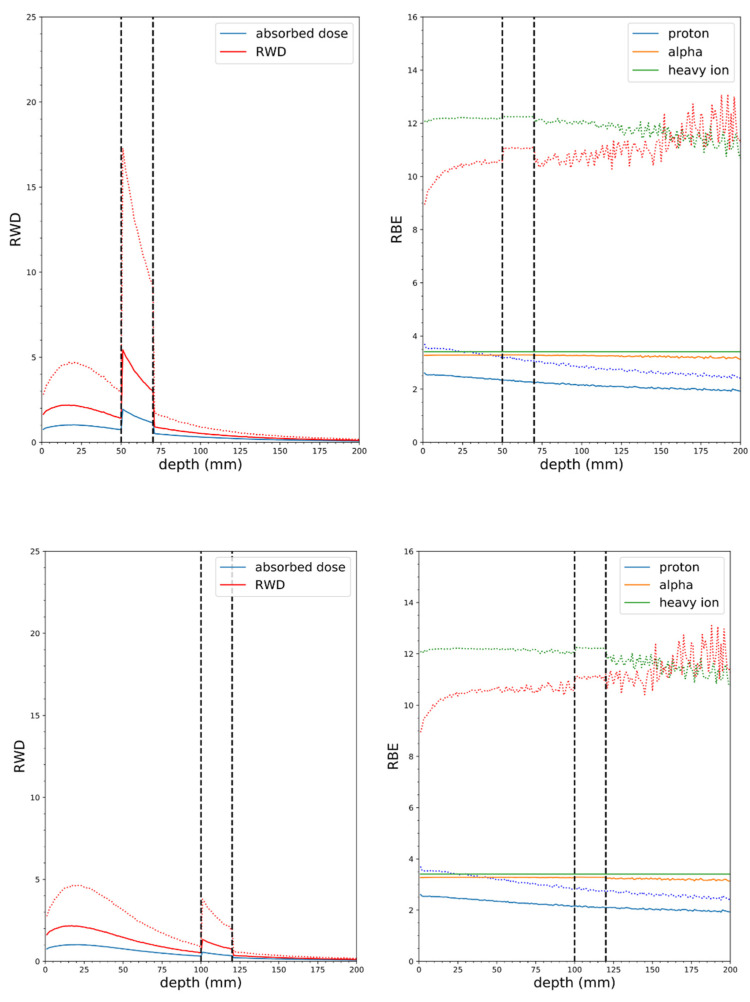
Soft tissue phantom simulations with NM neutron source, mass at 5 cm depth (top panel), 2 cm wide, 10:1 tumor to tissue ratio, 100 ppm ^10^B in tumor (vertical dashed lines indicate tumor borders), solid lines RBE(DSB), dotted lines RBE_LD_ (α/β = 3 Gy). Tumor at 10 cm depth (bottom panel).

**Figure 10 cells-09-02302-f010:**
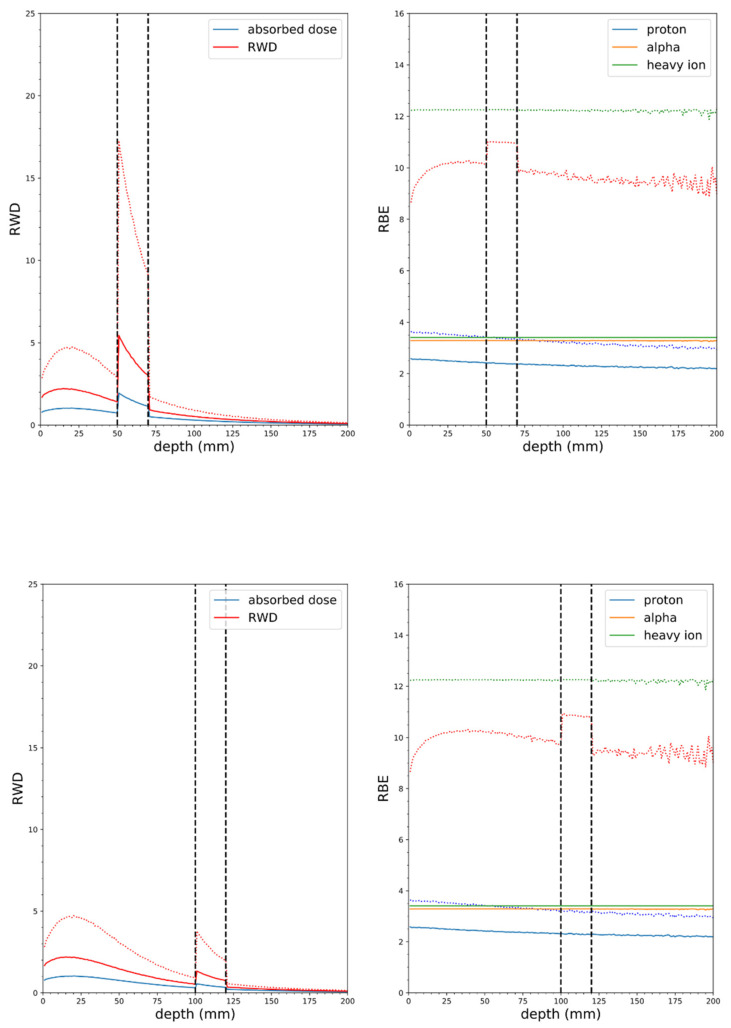
Soft tissue phantom simulations with CN neutron source, mass at 5 cm depth (top panel), 2 cm wide, 10:1 tumor to tissue ratio, 100 ppm ^10^B in tumor (vertical dashed lines indicate tumor borders), solid lines RBE(DSB), dotted lines RBE_LD_ (α/β = 3 Gy). Tumor at 10 cm depth (bottom panel).

**Figure 11 cells-09-02302-f011:**
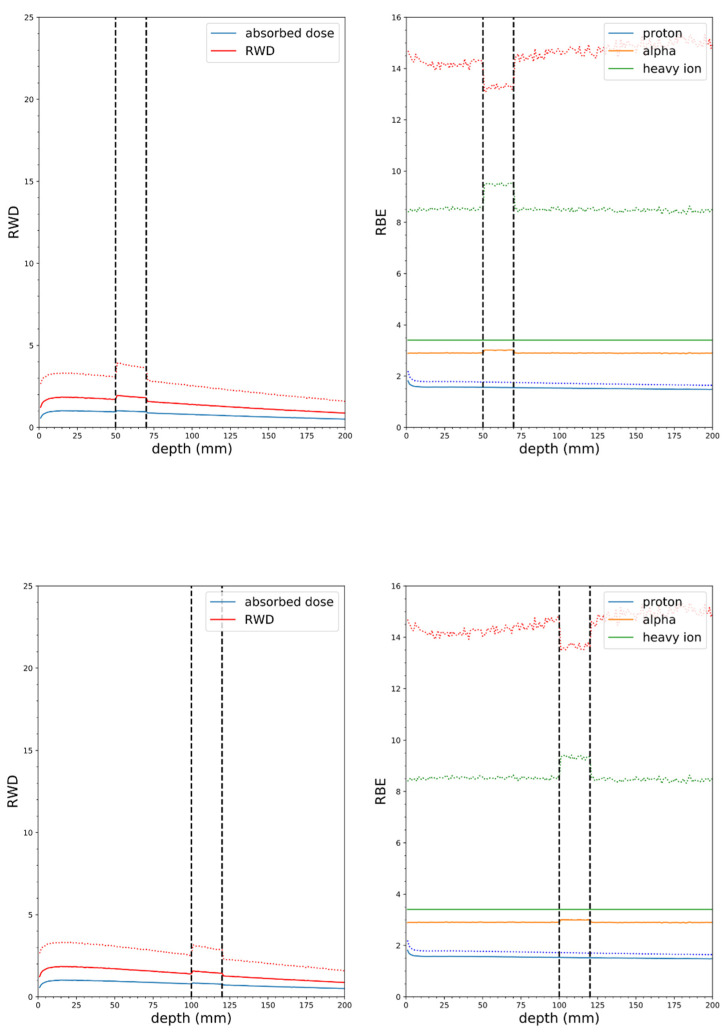
Soft tissue phantom simulations with CNTS neutron source, mass at 5 cm depth (top panel), 2 cm wide, 10:1 tumor to tissue ratio, 100 ppm ^10^B in tumor (vertical dashed lines indicate tumor borders), solid lines RBE(DSB), dotted lines RBE_LD_ (α/β = 3 Gy). Tumor at 10 cm depth (bottom panel).

**Table 1 cells-09-02302-t001:** LET, CSDA ranges and ${\bar{z}}_{F}$ calculations for selected ions computed using MCDS [32].

Particle	E (MeV)	LET (keV/μm)	CSDA Range (μm)	${\bar{z}}_{F}\;(\mathrm{Gy}){\;}^{a}$	${\bar{z}}_{F}\;(\mathrm{Gy}){\;}^{a}\;(\mathrm{ICRU}\;\mathrm{def}.){\;}^{b}$
^1^H^1+^	0.59	38.03	11.09	0.34	0.31
^4^He^2+^	1.47	186.5	8.28	1.69	1.52
^4^He^2+^	1.78	170.4	10.02	1.54	1.39
^7^Li^3+^	0.84	369.1	4.18	1.79	3.01
^7^Li^3+^	1.01	386.1	4.63	2.07	3.15

^a^ MCDS *d* = 5 μm, ^b^
${\bar{z}}_{F}$ = 0.204·LET/*ρd*^2^, *ρ* = 1 g/cm^3^.

**Table 2 cells-09-02302-t002:** Comparison of experimental results and model estimates for alpha particles and protons with approximately the same LET.

Particle	LET (keV/μm)		α (Gy^−1^)		RBE_DSB_
	Goodhead et al.	MCDS *	Goodhead et al.	MCDS + RMF *	MCDS *
1.2 MeV ^+^H	22.02	23.65	0.30	0.29	1.80
1.4 MeV ^+^H	19.67	21.13	0.42	0.27	1.71
30 MeV α	23.00	22.72	0.21	0.25	1.56
35 MeV α	20.45	20.07	0.25	0.23	1.50

* All Monte Carlo simulations run to standard error of the mean of 0.001.

**Table 3 cells-09-02302-t003:** Comparison of experimentally-derived parameters, MCDS + MCNP estimates and RMF fits.

	Dose-Weighted LET (keV/μm)		${\bar{z}}_{F}\;(\mathrm{Gy})$			RBE_DSB_
α Energy (MeV)	Tracy et al.	MCDS + MCNP	MCDS + MCNP		RMF Fit	MCDS + MCNP
			5 µm	3–6 µm		
1.1	181	203	1.09	3.76–0.67	1.03	3.30
1.5	201	213	1.60	5.33–1.16	1.66	3.24
1.8	190	195	1.69	4.67–1.16	2.68	3.19
2.4	161	161	1.44	3.84–1.02	3.32	3.09
3.2	131	130	1.13	3.06–0.79	4.21	2.96
4.0	112	110	0.94	2.56–0.66	4.90	2.84

**Table 4 cells-09-02302-t004:** Predicted RBE values for BNCT secondary charged particles using Equation (4).

	Protons		Alphas		Lithium		Heavy Ions	
Neutron source	RBE_HD_ = RBE_DSB_	RBE_LD_ (α/β = 3 Gy)	RBE_HD_ = RBE_DSB_	RBE_LD_ (α/β = 3 Gy)	RBE_HD_ = RBE_DSB_	RBE_LD_ (α/β = 3 Gy)	RBE_HD_ = RBE_DSB_	RBE_LD_ (α/β = 3 Gy)
MIT-FCB	2.85	3.42	3.06	8.79	3.39	7.05	3.15	6.39
NM	2.57	3.47	3.02	8.74	3.39	7.08	3.15	6.55
CN	2.54	3.38	3.04	9.06	3.39	6.82	3.15	5.59
UW CNTS	2.22	3.01	2.79	7.86	3.39	7.05	3.15	11.2

**Table 5 cells-09-02302-t005:** Dose-weighted RBE estimates for selected neutron sources and ^10^B carriers. Estimates are reported using asymptotic low and high-dose RBE models.

	RBE_HD_ = RBE_DSB_			RBE_LD_ (α/β = 87)			RBE_LD_ (α/β = 3)			RBE_LD_ (α/β = 10)
Neutron source	no ^10^B	BPA	mAb	no ^10^B	BPA	mAb	no ^10^B	BPA	mAb	no ^10^B
MIT-FCB	2.87	2.97	3.04	2.89	3.06	3.17	3.46	5.78	6.99	3.09
NM	2.63	2.66	2.69	2.68	2.71	2.75	3.99	4.08	4.34	3.02
CN	2.62	2.63	2.69	2.66	2.68	2.75	3.71	3.94	4.47	2.94
UW CNTS	2.66	2.66	2.72	2.80	2.81	2.88	6.86	6.88	7.40	4.03

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
