# Peer review of "Mechanistic Modeling of the Relative Biological Effectiveness of Boron Neutron Capture Therapy"

_cells, 2020, doi:10.3390/cells9102302_

Round 1

Reviewer 1 Report

The paper by Streitmatter et al. presents a multi-scale system of models which is supposed to evaluate the effectiveness of different neutron sources applied to Boron Neutron Capture Therapy (BNCT). This manuscript contains a detailed description of approaches verging on mathematics, physics and radiotherapy. For the readers without a solid background in these fields the content of this paper will be difficult to comprehend. “Cells” journal is associated with cellular biology, so the reader would expect that at least some experiments were performed on any kind of cell or animal model. This expectation is strengthened by the sentence from the Abstract (l. 20): “The multiscale model is tested against in vitro and in vivo measurements of cell survival with and without boron”.

In order to make it more accessible to those who are not familiar with non-biological models, I suggest to:

  1. modify the Abstract in such a way to make it clear that no biological model was used in this particular study and that the proposed model was validated based on already published data on cell survival.
  2. include a description of cell treatment, not only a reference to a paper, wherever applicable.
  3. include more details in the legends to the figures – a title is not enough.
  4. improve Fig. 1: I fully understand that it is not possible to use a real scale, but it is not clear which element of the model on the left panel is supposed to have 1 mm?

Minor:

The sentence in lines 395-398 is difficult to understand, could you please rewrite it?

Reviewer 2 Report

General comments:

The work is interesting and appears to have technical merit (so far as I am able to assess), but the paper is let down by the many significant defects in the presentation of the data - principally the lack of confidence intervals on the figures (and uncertainties in the tables). This is an absolutely critical element of the results that you are presenting. Without them, a reader cannot draw any conclusions on the data presented.

The presentation of the paper can (and must) be greatly improved. I recommend that you use LaTeX for manuscript preparation if this is an option. This will avoid several of the major formatting / presentation issues identified below (including incorrectly numbered tables and figures split across multiple pages). The use of vector graphics will also be a great improvement - low resolution bitmaps should never be used for graphs or line drawings.

There are many typographic and grammatical errors (to many to enumerate, although mostly these are in the first half of the paper). I recommended that the paper be thoroughly proof-read by a person with full professional proficiency in English prior to any resubmission.

Specific comments:

1. The discussion on microdosimetry (p3) is rather long and can probably be significantly condensed via reference to some of the major works in the field (even standard textbooks, e.g. Attix) as there is little value in rehashing this discussion in a research article. What is there is poorly referenced - for example, definitive statements about uncertainty in RBE estimates with no supporting citations.

RBE can be defined for any radiation field, it requires no 'generalisation' to be extended to a mixed radiation field as it is an experimental measurement. Note: there is no mention of the fact that RBE should be defined for a given cell survival probability (e.g. 10%) - despite nearly two pages of background on the concept.

2. Text in Figure 1 is completely illegible. Please use vector graphics for such figures.

3. From microdosimetry we jump into an algorithm (MCDS) which is never defined. What is it for? "It as been extensively tested" but we don't yet know its function. This makes the paper disjoint and difficult to read. Also, MCNP is introduced without the acronym being explained. You start talking about DE DF "cards" and some mysterious F6 - you are referring to punch cards? What century are we in? This should be explained or omitted (I don't think it is very relevant, it is an implementation detail and totally incomprehensible to readers unfamiliar with MCNP). Similarly RMF is not defined.

4. Neutrons are not ions. Do not refer to them as such.

5. Please define acronym CSDA

6. Please avoid vague language like "a bit". This is a scientific paper.

7. As the dimensions are unreadable on Figure 1, and the dimensions of the surrounding materials are not clearly described in the text, it is difficult to understand the simulation scenario. I think if I understand correctly you obtain phase space data for different depths in a macroscopic simulation, which you use to drive the microdosimetric simulations at each depth? Or is only the spectrum recorded at each depth and used to generate either an isotropic or directional neutron field for the second simulation? This is not clear.

8. From line 268, you mention that the boron distribution is based on some previous experimental studies. As this is a critical aspect of the present work, you need to be more specific about the boron concentrations (for uniform, provide the overall concentration in mol/L or ppm; for the non-uniform, explain the distribution and probably indicate the maximum/minimum or tumour:normal concentration ratios). Actually it seems this is mentioned in some of the figure captions (Figures 8-11) but it is critical to include it in the methodology (I couldn't find it - if it is there somewhere, it should be more prominent!).

9. The D-T neutron source is not shown in Figure 3. This would be helpful in understanding this NM model.

10. Line 377: what did you expect it to be? How big is the discrepancy?

11. Line 379: don't use the word "significant" without quantification please.

12. Table 2 (page 12) inexplicably precedes Table 1.

13. Table 2 (page 12) does not include any uncertainties. This is an important component of your results, without which their validity cannot be properly assessed.

14. Figure 6 omits confidence intervals - these are essential. The figure is also very hard to read (4.0 MeV is basically invisible) - again, use vectorised figures NOT bitmaps for line drawings or graphs. Yellow is a poor colour choice when printed on white.

15. There's another Table 2 on page 14, which also does not indicate any uncertainties.

16. Page 14: you are referring to Table 4, which does not appear to exist. Is it the second instance of Table 2? Or is that Table 3? If it is Table 2 number 2, then alpha/beta has a unit (Gy) which you didn't include.

17. Figure 7 has no confidence intervals.

18. No confidence intervals on Table 5. No uncertainty?

19. Figure 8-11 are all split over two pages. This makes the paper very hard to read and will be avoided if you use LaTeX.

20. Also no confidence intervals on any of Figures 8-11. This is essential and must be included.

Round 2

Reviewer 1 Report

The manuscript by Streitmatter et al. has been improved according to the reviewer’s suggestion, thank you for your effort.
